# The Caucasian Whortleberry Extract/Myrtle Essential Oil Loaded Active Films: Physicochemical Properties and Effects on Quality Parameters of Wrapped Turkey Breast Meat

**DOI:** 10.3390/foods11223553

**Published:** 2022-11-08

**Authors:** Vahid Bagheri, Babak Ghanbarzadeh, Karim Parastouei, Mohammad Hadi Baghersad

**Affiliations:** 1Health Research Centre, Life Style Institute, Baqiyatallah University of Medical Sciences, Tehran P.O. Box 51666-16471, Iran; 2Department of Food Science and Technology, Faculty of Agriculture, University of Tabriz, Tabriz P.O. Box 51666-16471, Iran; 3Department of Food Engineering, Faculty of Engineering, Near East University, 99138 Nicosia, Northern Cyprus, Turkey; 4Applied Biotechnology Research Center, Baqiyatallah University of Medical Sciences, Tehran P.O. Box 51666-16471, Iran

**Keywords:** active films, antimicrobial properties, physic-mechanical properties, turkey breast meat

## Abstract

In this research work, the effects of myrtle essential oil (MEO) and Caucasian whortleberry extract (CWE) as natural additives were investigated on mechanical, physico-mechanical and antimicrobial properties of gellan/polyvinyl alcohol (G/PVA) film. Then, optimal blend active films were used for the wrapping of turkey breast meat stored at low temperature (4 ± 1 °C) for 15 days and chemical and sensory properties of wrapped meats were evaluated. The addition of MEO and CWE decreased tensile strength and increased the strain at the break of the films (*p* ≤ 0.05). Additionally, with increasing the amount of MEO and CWE, the permeability to water vapor (WVP) and the moisture content (MC) of the films decreased (*p* ≤ 0.05). MIC test showed that MEO and CWE were effective against *S. aureus*, *E. coli*, *S. typhimurium*, and *P. fluorescens*. at the concentrations of 5–6 and 15–17 mg/mL, respectively. Different microbiological, chemical, and sensory tests indicated that active films significantly enhanced the shelf life of turkey breast meat (*p* ≤ 0.05). Therefore, based on our finding in this study, the use of these active and biodegradable packagings can be effective and useful for protecting the microbial and sensory quality of turkey breast meat.

## 1. Introduction

Food packaging has an important role in food shelf life and can delay food spoilage, extend shelf life, and retain the quality and safety of food. Therefore, packaging protects food from external chemical, biological and physical influences [1]. In recent years, increasing consumer demand for high-quality and safe food has led to the development of research into the production of the biopolymer-based biodegradable packaging films [2]. Up to now, biodegradable packaging films have been developed into two types; edible and non-edible biopolymers. Bio-materials such as polysaccharides, proteins, and lipids or a combination of them usually are used to produce biodegradable food packaging films [3]. Gellan gum is a microbial linear anionic heteropolysaccharide produced from aerobic fermentation of Pseudomonas elodea and has good resistance to heat and acid stress during production. In terms of functional properties, it is an excellent gelling agent that has similar functional properties to alginate, carrageenan, pectin, gelatin, and starch. It is also a biodegradable and non-toxic biopolymer. Additionally, gellan is an important and promising source for the production of bioplastics due to its excellent gel-forming properties, good film-forming properties, and rapid biodegradability [4]. However, pure gellan film, like many biodegradable films based on biopolymers, is hydrophilic and loses its mechanical and gas barrier properties due to hydration. Therefore, to solve this defect, the combination of gellan with other biodegradable biopolymers, hydrophobic compounds such as lipids and essential oils, and nanofillers can be used [4,5]. Polyvinyl alcohol (PVA) is a synthetic, biodegradable, and non-toxic polymer and is one of the suitable biodegradable polymers for improving the physical properties of polysaccharide films [6]. Polyvinyl alcohol (PVA) is widely used in combination with various biopolymers due to its high tensile strength, flexibility, and excellent film-forming properties, as well as its higher moisture and water vapor barrier properties than most polysaccharides and proteins [7]. In recent years, attentions have been increased on the advantages and application of essential oils (EOs) and herbal extracts, especially on their antimicrobial and antioxidant effects, which can be related to their extraction from natural resources, and having less risk to consumer’s health compared to chemical preservatives [8,9,10]. The essential oil of myrtle contains several major bioactive compounds such as 1,8-cineole, α-pinene, limonene, and linalool and some trace components [11,12]. The bioactive compounds of myrtle EOs can interact with food stuff such as meat and show antimicrobial and antioxidant effects, which potentially can delay microbial spoilage and lipid oxidation [11]. A member of the *Ericaceae* family, the Caucasian whortleberry (CW) or *Vaccinium arctostaphylos* L. is described in Iran as Qare-qat [13]. CW possesses antimicrobial and antioxidant effects because of the presence of phenolic and anthocyanin compounds [14]. Additionally, it is known as an important medicinal plant. In several studies, CW has been used for the treatment of infections, diabetics, and some other disorders [15,16].

The high levels of nutrients present in poultry meat make it an appropriate environment for the growth and proliferation of microorganisms, which increases the poultry meat spoilage rate and growth speed of harmful microorganisms in food [17,18]. The application of chemical preservatives in poultry meat products is one of the maintenance methods. However, the application of these chemical compounds in meat products and their side effects has increased consumers’ concern [19]. One of the new methods for increasing fresh poultry meat shelf life with minimal side effects includes non-thermal inactivation approaches such as active packaging (AP) and the use of naturally occurring antimicrobial agents. The main benefit of edible films as AP for application in food packaging is their suitable role as a carrier with the gradual and controlled release of preservative compounds [20]. To our knowledge, there is no study on gellan/PVA blend films containing myrtle essential oil and Caucasian whortleberry extract. The active packaging produced contains myrtle essential oil and Caucasian whortleberry extract containing phenolic compounds and chemical structures with lipophilic and hydrophobic functional groups that can lead to maintaining the quality and microbial and chemical properties of packaged turkey meat. The aim of this study was (1) to investigate the effect of myrtle essential oil and Caucasian whortleberry on some physicochemical and antimicrobial properties of gellan/PVA-based film and (2) to specify the effect of optimal antimicrobial film on the quality parameters of the wrapped turkey breast meat.

## 2. Materials and Methods

### 2.1. Materials

Gellan gum (low acyl) with a gel strength of 970 g/cm2 (0.5% gellan solution) was purchased from Foodchem International Co. (Shanghai, China). Polyvinyl alcohol with a 99 mol% hydrolysis degree, an average molecular weight of 89,000–98,000 Da was obtained from Sigma-Aldrich (St. Louis, MO, USA). MEO containing 50.96% α-pinene as the main chemical composition of MEO was purchased from Tabib Daru Co. (Kashan, Iran). CW was prepared from a local shop in the Aras baran region (Kaleybar, Iran). Violet Red Bile Dextrose Agar (VRBDA), Plate Count Agar (PCA), De Man, Rogosa and Sharpe Agar (Darmstadt, MRS), Mueller Hinton Agar (MHA), Salmonella Shigella Agar (SSA), Mueller Hinton Broth (MHB) and Eosin Methylene Blue (EMB) agar were all obtained from Merck Co. (Darmstadt, Germany) while Baird-Parker Agar (BPA) was purchased from Quelab (Montréal, QC, Canada) and Cephaloridine Fucidin Cetrimide (CFC) agar was provided from Micromedia (Montréal, QC, Canada). Moreover, glycerol was acquired from Merck Co. (Darmstadt, Germany). Additionally, all chemical materials used in this research with analytical grade were purchased from Merck co (Darmstadt, Germany). All bacterial strains used in the present study including *E. coli* O157:H7, *S. aureus*, *P. fluorescens,* and *S. typhimurium* were obtained from the Biological and Genetic Resources Center (Tehran, Iran).

### 2.2. Preparation of CWE

CW extract was prepared according to described method by Garofulic et al. with minor modification [21]. The fruits of CW obtained from a local market were screened and the impurities were removed. Then, the samples were washed and dried. The dried fruits (30 g) were well-milled and after mixing with 250 mL ethanol 70% were stirred at low speed for 48 h. In the following, the suspension filtered using Whatman paper No. 40 was then centrifuged for 5 min at 3000 rpm. The resulting suspension was concentrated to complete the removal of ethanol at 40 °C using a vacuum rotary evaporator (Heidolph, Germany). Finally, the concentrated extract was poured into a dark bottle and placed for fresh use over 5 days inside a freezer at −70 °C.

### 2.3. Preparation of G/PVA/MEO/CWE Films

The biodegradable active films were produced by a casting method. The formulations and composition of biodegradable active films are presented in Table 1. Gellan was dissolved in distilled water (50 mL) by stirring (200 rpm for 30 min) and heating (75 °C for 30 min). In the following, the PVA was gently stirred in distilled water (30 mL) at a set temperature of about 70 °C for 20 min using a magnetic stirrer. An emulsion solution containing antibacterial agents (MEO and CWE) was prepared using Tween 80 as an emulsifier in 20 mL of distilled water. Then, to obtain suitable dispersion, the emulsion solution was created by continuously stirring the solution in an ultrasonic homogenizer Model T25, Janke and Kunkel Ultra Turrax (Germany). After the addition of glycerol (30 wt%) as a plasticizer, the solution was mixed slowly (20 min) by a stirrer to eliminate the air bubbles. A certain volume (40 mL) of the film solution was poured into a Teflon mold (12 cm diameter) and dried at 35 °C in an incubator for 24 h [22]. A schematic of the process used to produce active films is presented in Figure 1.

### 2.4. Characterization of Films

#### 2.4.1. Mechanical Properties

Mechanical properties of biodegradable active film ultimate tensile strength (UTS) and percentage of strain at break (SB) were analyzed following a standard method of ASTM (Standard D638, 2010) on the conditioned films at 55 ± 3% RH and 25 °C for 24 h using a Sanaf Universal Testing Machine (Tehran, Iran) with a 25 N load cell and a crosshead speed of 5 mm/min at room temperature. The specimens were cut into dumbbell shapes with a special cuter and then loaded with an initial grip separation of 50 mm. Each specimen tested at least 3 replicates and their average was reported.

#### 2.4.2. Film Thickness Properties

The thickness of the active film samples was measured with a hand-held digital micrometer to the nearest 0.01 mm (GUANGLU, CHINA) at five random locations after 2 days of conditioning at 25 °C by 55 ± 3% relative humidity.

#### 2.4.3. Determination of Water Vapor Permeability (WVP)

The water vapor permeability (WVP) rate of the active films was calculated according to ASTM E96-95 standard method. All film samples were conditioned at 55 ± 3% RH at 25 °C for 24 h. Afterwards, each test cup containing 3 gr anhydrous calcium sulfate (0% RH) was covered with film samples. Then, all test cups were placed in a desiccator containing saturated potassium sulfate solution (97 ± 2% RH) at 25 °C. The water vapor transmission rate of active films was studied from the weight gain and absorbed cups over different times. After that, the slope was obtained by linear regression from the weight and time changes. Finally, the water vapor permeability (WVP) of active films was calculated from the following Equation (1): (1)$$\mathrm{wvp}=\frac{\mathrm{WVTR}.X}{P\left(R_{2}-R_{1}\right)}$$

Here, the water vapor transmission rate (WVTR) was defined as the slope (g/h) specified by the transfer area (m^2^). X is the average film thickness (mm). *p* is the saturated vapor pressure of water (Pa) at the test temperature (25 °C), R_2_ is the RH in a desiccator, and R_1_ is the amount of RH in the test cup. All measurements were carried out in least 3 replicates.

#### 2.4.4. Moisture Content (MC)

To measure the moisture content of biodegradable active films, each film sample was cut into 3 × 3 cm squares and placed on glass Petri dishes. Then, samples were dried in an oven at 105 °C for 24 h. By measuring the weight of the samples before and after drying, weight loss was measured as water content and, based on the initial weight of the film, was reported as a percentage [23].

#### 2.4.5. Field Emission Scanning Electron Microscopy (FE-SEM) 

The surface structure of the biodegradable active films was evaluated after coating them with a gold layer at an acceleration voltage of 10 kV using field emission scanning electron microscopy (Tescan, MIRA3, Brno, Czech Republic). All film samples used for testing before the test were conditioned for 2 days at 25 °C by 55 ± 3% relative humidity using a desiccator [3].

#### 2.4.6. Atomic Force Microscopy (AFM) 

Analysis of surface topography characteristics of the biodegradable films was performed by AFM (Nanosurf Mobile S, Liestal, Switzerland) with dynamic force mode and Tap190Al-G AFM probes at 25 °C. All film samples used for testing before the test were conditioned for 2 days at 25 °C by 55 ± 3% relative humidity using a desiccator. Sample data were converted to 3D images and were examined. Additionally, the surface roughness amount of the films was calculated using the Nanosurf software (Version 3.1.0.4, Nanosurf, Liestal, Switzerland) [22].

#### 2.4.7. Antibacterial Properties 

##### MIC and MBC Studies 

MIC and MBC tests were performed to determine the minimum inhibitory concentration (MIC) and the minimum bactericidal concentration (MBC) of MEO and CWE on 96-well microplates using the micro-dilution method based on the method described by Azizi et al. [24]. All bacterial strains (*P. fluorescens*, *S. typhimurium*, *S. aureus,* and *E. coli*) were cultured on Mueller hinton broth (MHB) for 24 h, and then adjusted to 0.5 McFarland standard turbidity (0.5 MFST) (1.5 × 10^8^ CFU/mL). In the following, various concentrations of CWE dispersed in distilled water, MEO with the same concentrations dispersed in dimethyl sulfoxide (DMSO), and a certain amount (20 μL) of the prepared bacterial suspensions were added to the MHB broth (160 μL) in the test wells. Additionally, to prepare a mixture of MEO and CWE, these compounds were mixed in a ratio of 1:1. Then, different concentrations of MEO/CWE were mixed with 160 μL of MHB containing 20 μL of the prepared bacterial suspensions in the test wells. Additionally, to confirm the MIC test results, positive controls (broth and antimicrobial materials containing MEO and CWE) and bacterial suspensions and broth as negative controls were used. Then, all of the microplates containing *E. coli*, *S. typhimurium*, and *S. aureus* were incubated for 24 h at 37 °C while the microplates containing *P. fluorescens* for 24 h at 25 °C were incubated. The lowest concentration without any observable bacterial growth as MIC values was specified. Additionally, based on MIC and MBC results, the suitable concentration of MEO and CWE to be added to edible films were selected.

##### The Antibacterial Activity Determination of G/PVA/MEO/CWE Films

The G/PVA/MEO/CWE film antibacterial activity was evaluated by the disc diffusion (DD) test. Firstly, 0.5 McFarland turbidity standards (MCF: 1.5 × 10^8^ CFU/mL) of *S. typhimurium*, *S. aureus*, *P. fluorescens*, and *E. coli* after 24 h proliferation at 37 °C in Mueller Hinton Broth (MHB) cultures were obtained. In the following, 100 μL of the suspensions of bacteria (1.5 × 10^8^ CFU/mL) were used for the inoculation of the Mueller Hinton Agar (MHA) surface. G/PVA/MEO/CWE films under sterile conditions sliced into a circle shape (10 mm diameter) were loaded on the surface of MHA in the cultivated plates. Other than *P. fluorescens* incubated at 25 °C, all other plates containing *E. coli*, *S. typhimurium* and *S. aureus* were incubated for 24 h at 37 °C. The inhibition zone around the discs was calculated using a digital micrometer. Finally, based on the DD results, the best G/PVA/MEO/CWE films to use in turkey breast meat samples packaging were selected [24].

##### Turkey Breast Meat Preparation and Treatments

The turkey breast meat preparation was performed according to the previously reported method [22]. Fresh turkey breast meat samples were purchased from a reputable store in Tabriz, Iran, and immediately placed in a laboratory refrigerator (4 °C). Then, 27 g of turkey breast meat samples were cut and divided into 2 groups as controls and treatment. The turkey breast meat samples as controls were wrapped in G/PVA films without MEO/CWE (Positive control) and wrapped with polyethylene film (Negative control) while the treated turkey meat samples were wrapped in G/PVA /MEO/CWE films. Finally, the controls and treated samples kept for 15 days at 4 °C were assessed on days 0, 3, 6, 9, 12, and 15 (three-day intervals) for determining the microbial, chemical, and sensory properties of turkey breast meat samples.

##### Microbiological Analysis of Turkey Breast Meat Samples

The microbiological analysis of turkey breast meat was performed according to the previously reported method [22,24]. The microbial properties of turkey breast meat samples were evaluated by mixing a certain amount (27 g) of turkey breast meat with sterile peptone water (300 mL and 0.1%). Then, the sample mixture was homogenized for 5 min by using a stomacher 400 (Seward Medical, London, UK). For microbial count, serial dilutions of various microbial suspensions (10^−1^−10^−8^) were prepared. On pre-prepared agar plates, 100 μL of each serial dilution was spread in the following. The microbial enumeration was performed as follows: The TVC was evaluated using PCA at 30 °C incubated for 24 h. *Pseudomonas* spp count by CFC at 25 °C incubated for 24 h was determined. Lactic acid bacteria (LAB) was enumerated using MRS agar plates for 24 h at 37 °C. *Enterobacteriaceae* count was specified using VRBDA after 24 h incubated at 37 °C. *S. typhimurium* was enumerated using SSA. Additionally, *S. aureus* count, determined using BPA incubated at 37 °C for 24 h, was obtained [25,26].

##### Inoculation of Turkey Breast Meat Samples by General Food-Borne Pathogenic Microorganisms 

The inoculation of turkey breast meat by general food-borne pathogenic microorganisms was performed according to the previously reported method [22,24]. For inoculating food-borne pathogenic bacteria, turkey breast meat samples were freshly used. First of all, turkey breast meat samples (27 g) were sliced and the resulting pieces were sterilized with ethanol solution (95% *v/v*). The suspensions of *P. fluorescens*, *E. coli*, *S. aureus*, and *S. typhimurium* bacteria containing 10^4^ CFU/g were considered for the inoculation of meat samples. In the following, the treated turkey breast meat samples were wrapped in G/PVA/MEO/CWE films while control samples were wrapped in G/PV films. All wrapped samples (control and treated) stored at 4 ± 1 °C for 15 days were maintained under aseptic conditions. Then, microbial enumeration of samples was performed at time intervals of 0, 3, 6, 9, 12, and 15. For the bacterial enumeration, various dilutions of the inoculated bacteria including *S. typhimurium*, *S. aureus*, *P. fluorescens*, and *E. coli* were prepared. Then, the prepared dilutions on SSA, BPA, CFC, and EMB were inoculated, respectively. Finally, bacterial counts incubated at 37 °C for 24 h were determined.

#### 2.4.8. Chemical Analyses of Turkey Breast Meat

Total volatile base nitrogen (TVB-N) content, thiobarbituric acid reactive substances (TBARS) value, and peroxide (PV) amount of turkey breast meat samples were determined based on the method described by Nozari et al. [27]. Additionally, to measure the pH of the samples, 10 g of the samples were mixed with 20 mL of distilled water and homogenized. The homogenized mixture was then placed at room temperature for 5 min. Then, the pH of the samples was measured using a pH meter (Metrohm pH meter model 780, Herisau, Switzerland).

#### 2.4.9. Sensory Assessment

The samples’ sensory properties were assessed by a 9-point hedonic scale to assess the color, odor, total acceptance, and texture of turkey breast meat by selecting as 10 experienced panelists graduate students from the Department of Food Science and Technology at the University of Tabriz, Tabriz, Iran. The scale points considered were as follows based on the described method by Azizi et al. [24]: excellent: 9, very good: 7–8, good: 4–6, and bad or unacceptable: 1–3.

#### 2.4.10. Statistical Analysis

All experiments in our study were performed in three replications. Completely randomized design (CRD) with analysis of variance (ANOVA) was used by SPSS software (version 24.0, IBM; Armonk, NY, USA). Duncan’s multiple range test (*p* ≤ 0.05) was applied for mean comparison and determining significant difference among samples. Additionally, microbial, chemical, and sensory evaluations of turkey meat samples using the two-way Turkey’s test were applied to perform multiple comparisons between the means of the samples to analyze the significant differences between the treatments.

## 3. Results and Discussion

### 3.1. Mechanical Properties

Evaluation of the mechanical properties of edible films is one of the most important tests to determine and measure the ability of biodegradable films to maintain the structural integrity of the wrapped food product [28,29]. The mechanical properties and thickness parameters of the G/PVA blend films containing different concentrations of MEO and CWE are presented in Table 2. The highest value of ultimate tensile strength (UTS) (21.12 Mpa) and the lowest value of strain at break (SAB) (28.42%) among all samples were related to the control sample, which could be due to the presence of extensive hydrogen bonds between the biopolymer chains. The formation of a large number of hydrogen bonds between the biopolymer chains can lead to high cohesiveness and low flexibility in edible films. The addition of MEO and CWE to the G/PVA blend film reduced the stiffness (decrease in UTS) and increased the flexibility (increase in SAB) of the films. This effect was intensified in the films at higher concentrations of MEO and CWE. The highest (20.86 Mpa) and lowest (9.18 Mpa) values of UTS for the active blend films were related to the sample containing 3 mg/mL of extract (G/PVA/CWE 3) and 9 mg/mL of MEO and CWE (G/PVA/MEO 9/CWE 9), respectively. The addition of MEO in comparison to the CWE, in equal concentrations, showed greater decreasing effect on UTS values. The addition of MEO at all concentrations (3, 6, and 9 mg/mL) significantly reduced the UTS values of the samples compared to the control film (*p* < 0.05). Meanwhile, CWE did not show a significant decreasing effect at all concentrations (*p* > 0.05). This shows that MEO has more plasticizing effect than CWE which could be related to higher low molecular weight hydrophobic compounds in essential oil composition. The highest (76.18%) and lowest (30.65%) values of SAB for the active blend films were related to the sample containing 9 mg/mL of MEO and CWE (G/PVA/MEO 9/CWE 9) and 3 mg/mL of CWE (G/PVA/CWE 3), respectively. It seems that due to the hydrophilic nature of gellan and PVA, the addition of essential oil (MEO) and extract (CWE) reduced the structural cohesion of the films by decreasing hydrogen bonds and creating free spaces between the polymer chains. Therefore, films containing MEO and CWE showed less stiffness (lower UTS) and more flexibility (more SAB) compared to control films. Additionally, the thickness changes in most samples were not significant compared to the control sample. Nevertheless, the thickness of the sample containing the maximum amounts of MEO and CWE (MEO 9 mg/mL and CWE 9 mg/mL), compared to the control sample, showed a significant increase (*p* < 0.05) which could be attributed to decrease in compactness in film structure. Similar study results were reported for PVA-starch film containing lemongrass oil. Thus, the addition of lemongrass oil has reduced the amount of tensile strength and increased the value of strain at break [30]. Additionally, the addition of thymus zygis essential oil to the films based on gellan-starch reduced the tensile strength of the edible films [31].

### 3.2. Water Vapor Permeability (WVP) and Moisture Content (MC) Properties

The water vapor barrier properties of biodegradable films are one of the most important parameters in terms of preventing the absorption of moisture from the environment and loss of moisture which leads to maintaining and improving the quality, safety, and shelf life of the packaged foods [32]. Therefore, low permeability to water vapor is one of the essential features of food packaging films. The WVP and MC values of the G/PVA blend films containing different concentrations of MEO and CWE are presented in Table 3. The study of the obtained results showed that the addition of MEO and CWE could reduce the values of WVP in most of the samples compared to the control one. With increasing concentrations of MEO and CWE, the WVP values of films were more decreased. The lowest (0.79 × 10^−10^ g/mhpa) and highest (4.76 × 10^−10^ g/mhpa) values of WVP for active blend films were found in the sample containing the maximum amounts of MEO and CWE together (MEO 9 mg/mL and CWE 9 mg/mL) and 3 mg/mL of CWE (G/PVA/CWE 3), respectively. The higher number of water barrier effects of MEO can be attributed to the integration of oil with the biopolymer matrix, increasing the hydrophobic nature of biopolymers and reducing the affinity of the emulsified films to water molecules. Additionally, the study of the MC results showed that the G/PVA film containing the essential oil had lower moisture content than the control film (*p* < 0.05). The addition of the CWE at some concentrations also caused a decreasing effect in MC, however it has lower effect in comparison to MEO in the film matrix which could be attributed to lower hydrophobicity or compatibility of its chemical components. The lowest (2.51 %) and highest (10.48 %) values of MC for active blend films were related to the sample containing the maximum amounts of MEO and CWE together (MEO 9 mg/mL and CWE 9 mg/mL) and 3 mg/mL of CWE (G/PVA/CWE 3), respectively. In general, adding essential oil to the film substrate reduces the permeability to water vapor and thus causes relative hydrophobicity [31]. Additionally, the hydrogen interactions between the essential oil and the extract with polyvinyl alcohol and gellan reduce the hydrogen groups of the biopolymers, which can reduce the affinity of the films for water [30].

### 3.3. Surface Morphology of Biodegradable Films Using FE-SEM 

The morphology and surface structure images of G/PVA (a), G/PVA/CWE 9 (b), G/PVA/MEO 9 (c), and G/ PVA/CWE 9/MEO 9 (d) films, obtained using scanning electron microscopy, are shown in Figure 2. FE-SEM images of G/PVA films without MEO and CWE (as control film) showed a relatively smooth and homogeneous surface without pores or cracks. Additionally, the SEM images showed that the addition of CWE and MEO (9 mg/mL film solution) to the G/PVA matrix led to a relatively heterogeneous surface structure and a slight irregularity in the film surface. However, essential oil droplets were uniformly distributed on the film surface and accumulation of them was not observed. Co-addition of both components (MEO and CWE) to the G/PVA film samples at the highest concentration of MEO and CWE (9 mg/mL and 9 mg/mL) reduced the smoothness and homogeneity of the film surface. On the other hand, it increased the surface roughness and protrusion of the G/PVA/MEO/CWE film. Additionally, it seems that the reason for the increase in irregularity and heterogeneity of the film surface structure is due to the increase in the concentration of essential oil and extract and the presence of MEO and CWE droplet and particles on the film surface. The addition of MEO and CWE (9 mg/mL and 9 mg/mL) to the G/PVA film matrix had no significant negative impact, such as creating cracks, holes, or reduced film cohesion on the surface morphology of emulsified films. The reason for this can be attributed to the uniform distribution of MEO and CWE, suitable stability, and compatibility with the G/PVA matrix. Pirnia et al. [33] reported that pure gelatin-frankincense films had a smoother and more uniform surface morphology. Meanwhile, the addition of essential oil reduced the smoothness and uniformity of the gelatin-frankincense films. Additionally, the obtained results are in good agreement with a report of Wang et al. [34].

### 3.4. Atomic Force Microscope (AFM) 

The morphological properties of the biodegradable films were further investigated by AFM and surface topography images (qualitative parameter) and surface roughness values (quantitative parameter) of the biodegradable films were evaluated [3]. The AFM analysis (3D images and surface roughness values) are presented in Figure 3. The obtained results show that the G/PVA biodegradable films without MEO and CWE had low roughness, and smooth and uniform surfaces. The average roughness (Sa) = 9.25 nm and root mean square (Sq) = 18.02 nm for this sample were obtained. It is also observed that the addition of CWE (9 mg/mL) to the G/PVA matrix reduced the surface uniformity and increased the film roughness (Sa = 19.90 nm and Sq = 27.60). The study of 3D images and surface roughness values of the G/PVA biodegradable films containing MEO (9 mg/mL) showed that the addition of MEO can significantly decrease the surface smoothness and increase the surface roughness (Sa = 41.25 nm and Sq = 52.68). However, the biodegradable films containing MEO and CWE (9 mg/mL and 9 mg/mL) had Sa =80.39 nm and Sq = 91.21 nm, indicating that co-adding of MEO and CWE led to distribution on the biodegradable film surface. Therefore, the roughness of the film surface in this sample showed a significant increase compared to previous samples. Javidi et al. [35] also showed that adding essential oil to a film based on poly lactic acid significantly increases the surface roughness of the films, which can be associated with the formation and accumulation of lipids and thus causes irregularities in the surface of the films. Additionally, the results obtained from the present study are in good agreement with the results of the study by Dolatkhah et al. [22].

### 3.5. The MIC and MBC Evaluation for MEO and CWE

The MIC and MBC values of MEO and CWE are presented in Table 4. MIC test showed that MEO and CWE were effective against *S. aureus*, *E. coli*, *S. typhimurium*, and *P. fluorescens*. at the concentrations of 5–6 and 15–17 mg/mL, respectively. Additionally, MBC test indicated that MEO and CWE were effective against *S. aureus*, *E. coli*, *S. typhimurium*, and *P. fluorescens*. at the concentrations of 6–7 and 18–24 mg/mL, respectively. The MIC and MBC results in this research are in agreement with the previously published reports [24,36,37,38,39,40]. Based on the results, antimicrobial agents of MEO and CWE had suitable growth inhibition on bacteria. Additionally, in the concentration of 15 mg/mL, CWE had excellent activity against *E. coli* showing that *E. coli* was a more sensitive bacterium compared to other bacteria against CWE. Additionally, at a concentration of 3 mg/mL, the MEO had an excellent growth inhibition effect on *E. coli*, *S. typhimurium*, and *P. fluorescens*. G^+^ bacteria have multilayer peptidoglycan in their cell walls while G^–^ bacteria possess a thin peptidoglycan layer facilitating the effect of MEO and CWE on the bacteria cell walls. Therefore, G^+^ bacteria had more resistance than G^–^ bacteria. Finally, based on the obtained MIC results, 14 mg/mL of MEO/CWE (Mix both combinations at a ratio of 1:1) were considered as optimum concentrations in the preparation of active films for turkey meat packaging.

### 3.6. Antibacterial Activity of G/PVA /MEO/CWE Films

The antimicrobial activity of G/PVA/MEO/CWE films was determined using the disk diffusion (DD) method and the results are shown in Table 5. The DD results show that the G/PVA/MEO/CWE edible films had a suitable antimicrobial effect against G^–^ bacteria, especially against *E. coli*. Furthermore, to investigate the possible antimicrobial effects of the control samples, the G/PVA edible films as a positive control sample (without the MEO and CWE) and polyethylene film as a negative control sample were used. The positive and negative control samples had no inhibition effects on the evaluated bacteria in this research. In all active film samples, the highest and lowest inhibition zones were related to *E. coli* and *S. aureus*, respectively (*p* ≤ 0.05). Thus, the antibacterial activities of MEO and CWE might be attributed to the following reasons: The antibacterial agents such as MEO and CWE bind to the components of bacteria cells and interrupt the cytoplasmic membrane. Additionally, antibacterial agents change the permeability of the bacteria cell wall and affect the vital biomacromolecules, such as proteins and DNA [37]. Therefore, it can be said that the antimicrobial activity of MEO and CWE is related to the presence of phenolic compounds and chemical structures with lipophilic and hydrophobic functional groups. Besides, the antimicrobial activity of MEO and CWE against some bacteria by other researchers were provided as well. These results obtained here are in good agreement with the research previously reported up to now [38,41]. 

### 3.7. Microbial Analysis of Samples

For microbial analysis, the changes in the TVC, *Enterobacteriaceae*, *Pseudomonas* spp., and LAB counts in the raw turkey breast meat during refrigeration for 15 days were evaluated. Additionally, microbial analysis was performed on samples wrapped with G/PVA/MEO/CWE active films (treatment), wrapped with PE films as the negative control (Control 1), and wrapped with G/PVA films as the positive control (Control 2).

#### 3.7.1. Determination of TVC Values 

The TVC amounts of turkey breast meat wrapped in the G/PVA/MEO/CWE films as treated samples, wrapped in the PE films as the negative control (Control 1) and G/PVA films as the positive control (Control 2) are shown in Figure 4a. The initial amount of TVC for fresh turkey breast meat was obtained as 3.9 log CFU/g, which indicated that the quality of fresh turkey breast meat was good and suitable. The maximum acceptable and recommended population amount of TVC for the turkey breast meat was equal to 7 log CFU/g. The TVC population for turkey breast meat samples wrapped in G/PVA/MEO/CWE films during 15 days of storage at 4 °C were considerably lower than the TVC population of the negative control (Control 1) and positive control (Control 2) samples (Figure 4a). The TVC values of the wrapped turkey breast meat with the active film on days 9, 12, and 15 were achieved to be 5.1, 5.7, and 6.2 log CFU/g (*p* ≤ 0.05), respectively. However, the TVC population for the positive control samples was 7.1, 7.6, and 7.9 log CFU/g (*p* ≤ 0.05) on days 9, 12, and 15, respectively. Additionally, the TVC values of the wrapped turkey breast meat with the PE film as a negative control on days 9, 12, and 15 were achieved to be 7.4, 7.9, and 8.4 log CFU/g, respectively. The results obtained for TVC indicate that the application of G/PVA/MEO/CWE films for turkey breast meat packaging had significantly reduced the growth of microbes and increased the shelf life of turkey breast meat specimens. The antibacterial effects of CWE and MEO against both G^–^ and G^+^ bacteria have been presented in previous studies. Brink et al. [42] have reported that chitosan and whey protein films containing cranberry extract have reduced TVC counts in fresh cut turkey meat. Additionally, Dong et al. [43] indicated that the utilization of PE film containing rosemary and cinnamon essential oil can result in a controlled and decreased TVC population in white shrimp.

#### 3.7.2. *Pseudomonas* spp. Count

*Pseudomonas* spp. is recognized as completely aerobic G^–^ bacterium that is extremely susceptible to carbon dioxide. Besides, *Pseudomonas* spp. is one of the primary and important psychrotrophic microorganisms and is effective in the spoilage of turkey breast meat [44]. The initial population of *Pseudomonas* spp. for all specimens was 2.9 log CFU/g, but the population in the positive and negative controls samples after 15 days of storage reached 6.2 and 6.5 log CFU/g, respectively, as shown in Figure 4b. However, such an increasing trend was not observed for the treated samples. Therefore, on day 15, the population of *Pseudomonas* spp. for the treated samples with active films was equal to 4.6 log CFU/g. Thus, the G/PVA/MEO/CWE film can reduce the *Pseudomonas* spp. population compared to control samples. The reason for maintaining the antimicrobial effectiveness of the active films in the last days of storage can be attributed to the controlled release of the antimicrobial agents from the active film in the treated specimens. Raeisi et al. [9] indicated that the growth of *Pseudomonas* spp. was limited by sodium alginate containing rosemary EOs, nisin, and cinnamon essential oil in chicken meat stored at 4 °C. Additionally, Kakaei et al. [45] reported that the chitosan-gelatin containing red grape seed extract and Ziziphora clinopodioides essential oil show a good antimicrobial effect on psychrophilic microorganisms in the minced trout fillet. 

#### 3.7.3. *Enterobacteriaceae* Count

To measure the observance of food hygiene and environmental hygiene principles, *Enterobacteriaceae* bacteria could be considered as a good indicator [46]. 

The *Enterobacteriaceae* amounts of turkey breast meat wrapped in the G/PVA/MEO/CWE films as treated samples, wrapped in the PE films as negative control (Control 1) and G/PVA films as positive control (Control 2) are shown in Figure 4c. The initial population of *Enterobacteriaceae* on the first day (day 0) of storage in all turkey breast meat samples was equal to 1.3 log CFU / g, which indicated the good and suitable quality of turkey breast samples. The *Enterobacteriaceae* population increased during the storage of turkey breast meat samples. However, the control samples compared to the treated sample showed a higher population growth rate. Therefore, the *Enterobacteriaceae* population for positive control samples on the 3rd, 9th, and 15th days reached 2.3, 5.7, and 7.2 log CFU/g, respectively (*p* ≤ 0.05). Additionally, the *Enterobacteriaceae* values of the wrapped turkey breast meat with the PE film as a negative control on days 3, 9, and 15 were achieved to be 2.8, 5.9, and 7.8 log CFU/g, respectively. However, this amount in the treated samples on days 3, 9, and 15 were 1.5, 4.1, and 5.4 log CFU/g, respectively (*p* ≤ 0.05). So, the population of the control samples was higher than the treated samples for all days. The results obtained in this study are in agreement with the results acquired in a prior report [6,24,25]. The authors indicated that the utilization of PE films containing rosemary and cinnamon essential oil has considerably decreased the development of *Enterobacteriaceae* compared to the control sample stored at 4 °C [43]. The results of this study reveal that the population of Enterobacteriaceae in the treated samples increased with a gentle slope compared to the control samples. Therefore, the G/PVA/MEO/CWE films can be effective in reducing the slope of *Enterobacteriaceae* growth, and therefore, increasing the shelf life of turkey meat. Additionally, similar results were provided by Nirmal and Benjakul [47], who found that the active films containing green tea extract reduced the population of *Enterobacteriaceae* in the control sample compared to the treated samples throughout storage time at a temperature of 4 °C.

#### 3.7.4. LAB

The LABs (Lactic acid bacteria) are known as G^+^ (gram-positive) bacteria that are capable of growing under anaerobic to low aerobic conditions. LAB also plays a significant role in the normal micro-flora of turkey breast meat. On the first day (day 0), the Lactic acid bacteria population was equal to 1.8 log CFU/g. Although, as shown in Figure 4d, during the storage period, this number steadily increased. Based on the results obtained in this study, G/PVA/MEO/CWE films compared to the control samples can be effective in reducing the LAB growth slope on all days (3, 6, 9, 12 and 15), especially on the 15th day of storage. Therefore, the LAB population compared to negative control samples (Control 1) for the treated samples was reduced to about 1.4 log CFU/g at the end of 15 days’ storage time (*p* ≤ 0.05). Additionally, the LAB population compared to positive control samples (Control 2) for the treated samples was reduced to about 1.5 log CFU/g at the end of 15 days’ storage time (*p* ≤ 0.05). It can be seen that the LAB population in the treated samples was 5.8 log CFU/g at the end of the storage period, which has been significantly reduced in comparison with the negative control (7.3 log CFU/g) and positive control (7.2 log CFU/g) samples. These findings are consistent with the results presented by Kakaei et al. [45] who indicated a significant decrease in LAB in wrapped minced trout fillet with active films and stored at 4 °C. Another study reported that chitosan-essential oil nano-formulation on beef meat can limit the growth of LAB during storage at 4 °C [48].

### 3.8. Evaluating the Inoculated Foodborne Pathogenic Microbes 

Some pathogenic microorganisms such as *S. typhimurium*, *S. aureus*, *E. coli* and *P. fluorescens* are the most significant food-borne pathogenic bacteria that have important effects on turkey breast meat hygiene [49]. The antimicrobial effects of G/PVA/MEO/CWE films against *E. coli*, *S. typhimurium*, *S. aureus*, and *P. fluorescens* inoculated in turkey breast meat and stored at 4 °C are presented in Figure 5a–d, respectively. The bacteria population on day 0 for all the specimens was about 4 log CFU/g. The population for *E. coli*, *S. typhimurium*, *S. aureus* and *P. fluorescens* bacteria in negative control samples (Control 1) on the 15th day reached 7.3, 8.2, 7.9 and 7.7 log CFU/g, respectively (*p* ≤ 0.05). However, the growth slope obtained for *E. coli*, *S. typhimurium*, *S. aureus* and *P. fluorescens* in positive control samples (Control 2) at the 15 days of maintenance reached 7, 7.8, 7.7, and 7.3 log CFU/g, respectively, which was higher than the treated samples with active films at 15 days of maintenance. Meanwhile, the population of *E. coli*, *S. typhimurium*, *S. aureus*, and *P. fluorescens* bacteria reached 4.8, 5.8, 6.5 and 5.6 log CFU/g, respectively, in the treated samples with active films. Therefore, G/PVA/MEO/CWE films have considerable antimicrobial effects on harmful bacteria studied, principally against G^–^ bacteria. These results are well confirmed by the results achieved in the research reported by Yin Lau et al. [50] which had shown that the use of *Vaccinium macrocarpon* had a stronger inhibitory effect against pathogenic bacteria. Additionally, Ahmed et al. [51] described comparable results on the inhibitory effects of essential oil against pathogenic bacteria in chicken meat. The obtained results of the present study for psychrophilic bacteria population are in good agreement with the reports provided by other researchers [52].

### 3.9. Chemical Analysis on Wrapped Turkey Breast Meat

For chemical analysis, the changes in the pH, TBARS, TVB-N, and PV in the raw turkey breast meat during 15 days of refrigeration were investigated. Additionally, chemical analysis was performed on samples wrapped with G/PVA/MEO/CWE active films (treatment), wrapped with PE films as negative control (Control 1) and wrapped with G/PVA films as positive control (Control 2).

#### 3.9.1. Changes in the pH Values 

The changes in pH value of turkey breast meat during refrigerated storage (4 °C) are presented in Figure 6a. The initial pH of the turkey breast meat was 6.48. The pH values of turkey breast meat samples wrapped with active films did not significantly change during 15 days, however, the samples wrapped with the negative control (Control 1) and positive control (Control 2) showed the lowest pH value on the 15th day (*p* < 0.05) which may be attributed to higher bacterial spoilage. Therefore, the active packaging film containing antimicrobial compounds (MEO and CWE) was able to better maintain the pH of turkey breast samples than control samples due to having inhibitory effect on microbial growth. The obtained results of the present study for pH changes are in good agreement with the reports provided by other researchers [53,54].

#### 3.9.2. Changes in the TBARS 

The decomposition of hydroperoxides into secondary products of oxidation results in the formation of a wide range of compounds (such as carbonyls, hydrocarbons, furans, etc.) that contribute to creating spoilage and undesirable taste in food. Measuring the amount of thiobarbituric acid reactive substances (TBARS) is often used as one of the important indicators to determine the degree of lipid oxidation in foods. The effects of MEO- and CWE-loaded active packaging films based on G/PVA on the TBARS values of turkey breasts compared to the negative control (Control 1) and positive control (Control 2)) are shown in Figure 6b. Acceptable amounts of TBARS for turkey breast meat are less than 2 mg of MDA/kg (mg malondialdehyde / kg turkey breast sample). Additionally, TBARS values less than 1 mg of MDA/kg indicate that the turkey breast meat sample is fresh in terms of lipid rancidity. The initial amount of TBARS in the fresh turkey breast sample was 0.34 mg of MDA/kg of turkey breast meat. However, with increasing storage time, this value increased for all samples, but the amount of this increase was less in the sample wrapped with active film. The TBARS values of the samples wrapped with active film and the positive and negative control samples on the 15th day were 1.95, 3.62, and 3.84 mg of MDA/kg of turkey breast meat, respectively. Decreased oxygen availability and permeability and inhibition of the production of reactive oxygen species can be the main reason for the decrease in the TBARS values of the wrapped sample with active film. The results obtained by Sogut et al. [53] showed that the use of grape seed extract had a stronger inhibitory effect against increasing in the TBARS amounts of the wrapped chicken breast fillets. Additionally, similar results were reported by other researchers that were in good agreement with the results of the present study [27,45].

#### 3.9.3. Changes in the TVB-N 

The amount of TVB-N is often used as an important indicator to detect deterioration and protein and amine degradation in meat [27]. Therefore, high TVB-N values in meat are considered undesirable and poultry meat and products with TVB-N values of more than 25 mg/100 g are considered as spoiled and are not consumable by humans. The effects of the MEO- and CWE-loaded G/PVA biodegradable films on the total volatile basic nitrogen (TVB-N) changes of the wrapped turkey breast meat are shown in Figure 6c. The initial value of TVB-N for turkey breast meat was 4 mg N/100 g. However, the TVB-N value of the control samples increased faster than the treated sample with active films. The TVB-N values of the samples wrapped with active films and the negative control (Control 1) and positive control (Control 2) during 15 days of storage reached 21.4, 39.8, 39.2, and mg N/100 g, respectively (*p* < 0.05). The sharp and significant increase in the TVB-N value of the control samples on day 15 might be due to the increase in deterioration and protein and amine degradation by bacterial spoilage in the turkey meat. Additionally, the low TVB-N value of the treated sample on day 15 might be due to the effect of the antimicrobial compounds (MEO and CWE) with an inhibitory effect on microbial growth. Dolat khah et al. [22] reported that the initial TVB-N value of the chicken fillets was 6.12, which reached 33.15 mg/100 g after 12 days of storage at 4 °C.

#### 3.9.4. Peroxide Value Measurement 

Peroxide test was used to measure the amount of hydroperoxide (as primary products of lipid autoxidation) and the effects of the active packaging films based on G/PVA, containing optimal amounts of MEO and CWE, on the peroxide values of turkey breasts are shown in Figure 6d. The control samples (wrapped with PE films as negative control (Control 1) and wrapped with G/PVA films as positive control (Control 2)) were used for PV value evaluation. The initial amount of peroxide in the fresh turkey breast sample was 0 meq peroxide/1000 g lipids and it increased with increasing storage time. The samples wrapped with the control films showed a greater increase in PV compared to ones wrapped with the active films. The peroxide values of the sample wrapped with G/PVA/MEO/CWE active films and the negative and positive control samples on the 15th day were 1.20, 2.43, and 2.35 meq peroxide/1000 g lipid, respectively. Therefore, the results show that lower oxidative rancidity occurred in the sample wrapped with active packing film compared to the control ones. It seems that the presence of antioxidant compounds, antimicrobial compounds (decrease in hydrolysis rate) as well as reduced access to oxygen are the three main factors which had a reducing effect on the amount of peroxide in the samples wrapped with active film. Kakaei et al. [45] reported that active films containing red grape seed extract and Ziziphora clinopodioides essential oil can limit the increase in the peroxide value in the wrapped minced trout fillet during storage at 4 °C. Additionally, the results of other researchers were in good agreement with our findings in the present study [22,27,45].

### 3.10. Sensory Evaluation

The effect of G/PVA/MEO/CWE films on the sensory properties of turkey breast meat specimens (treated and control groups) are shown in Table 6. For all specimens, the sensory properties on the first day (day 0) of the experiments were desirable while desirability and acceptance of sensory characteristics during the storage time were decreased (*p* ≤ 0.05). The results of the sensory test indicate that until the 6th day of storage, all sensory properties (odor, color, texture, and overall acceptability) of the control specimens were appropriate. Meanwhile, the acceptability of these features extended up to day 15 of storage in the wrapped samples with active films (treatment). According to statistical analysis, a considerable difference (*p* < 0.05) was observed in terms of overall acceptance between the control and treated samples on days 3, 6, 9, 12, and 15 of the storage period. Our findings for active films containing MEO and CWE are in agreement with the results of Haute et al. [55] and Ghabraie et al. [56]. They indicate that the essential oil could increase the sensory properties of ground meat, fish and meat products throughout the storage time.

## 4. Conclusions

The main purpose of this study was to investigate the effects of MEO and CWE as natural compounds on the physicochemical properties of biodegradable films and also to evaluate their effects on microbial, chemical, and sensory properties of wrapped turkey breast meat samples. Therefore, the optimal amounts of antimicrobial compounds (MEO/CWE) incorporated in the film were determined by performing MIC and disc diffusion tests to achieve high inhibitory effects on microorganisms and also prevent negative effects on the sensorial properties of turkey breast meat. According to the results, it could be concluded that the use of G/PVA/MEO/CWE films can increase turkey breast meat shelf life significantly compared to control samples. Based on the sensorial, chemical, and microbial analyses, the maximum shelf life of control samples was only 6 days, but this amount was increased to 15 days in most samples wrapped with the G/PVA/MEO/CWE active films. Therefore, based on the positive results and the desired effect of the active films, the amount of 14 mg/mL of essential oil and extract was selected as the optimal amount. The application of MEO and CWE as a natural and novel preservative agent alone or in combination with other preservatives agents could be considered as a new approach in the development of active films for increasing the shelf life of turkey meat and meat-derived products.

## Figures and Tables

**Figure 1 foods-11-03553-f001:**
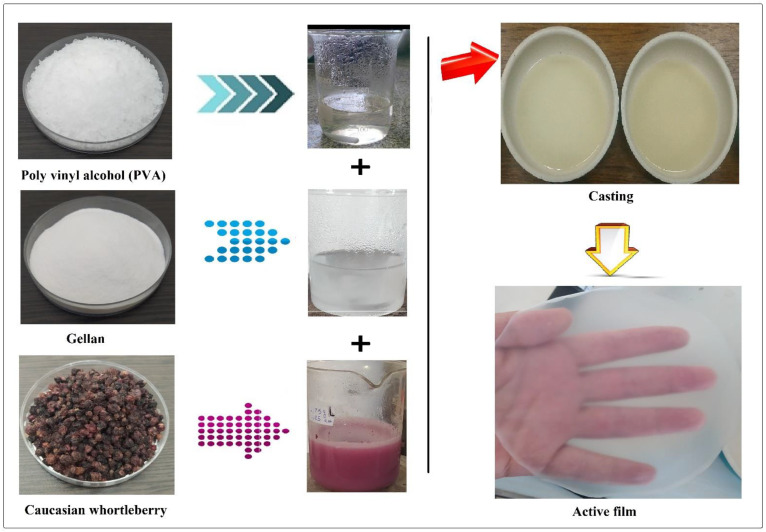
Schematic of the process used to produce the active films.

**Figure 2 foods-11-03553-f002:**
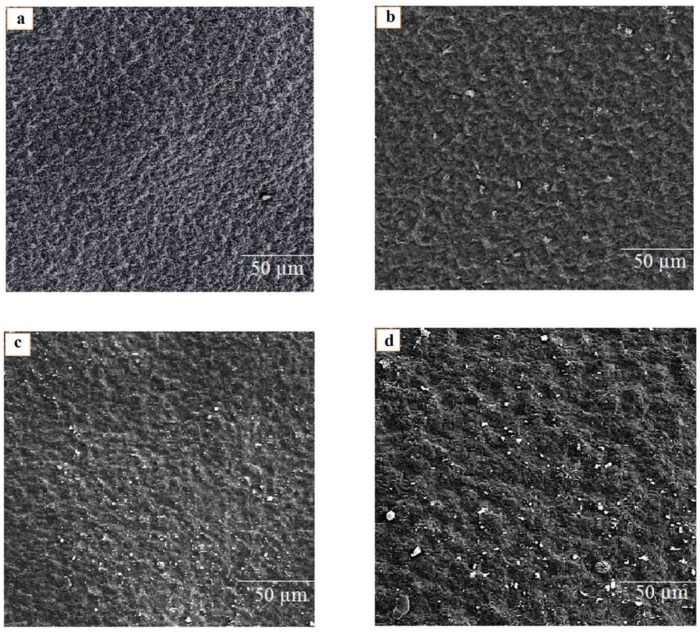
FE-SEM images of G/PVA (**a**), G/PVA/CWE 9 (**b**), G/PVA/MEO 9 (**c**), and G/ PVA/CWE 9/MEO 9 (**d**) active films.

**Figure 3 foods-11-03553-f003:**
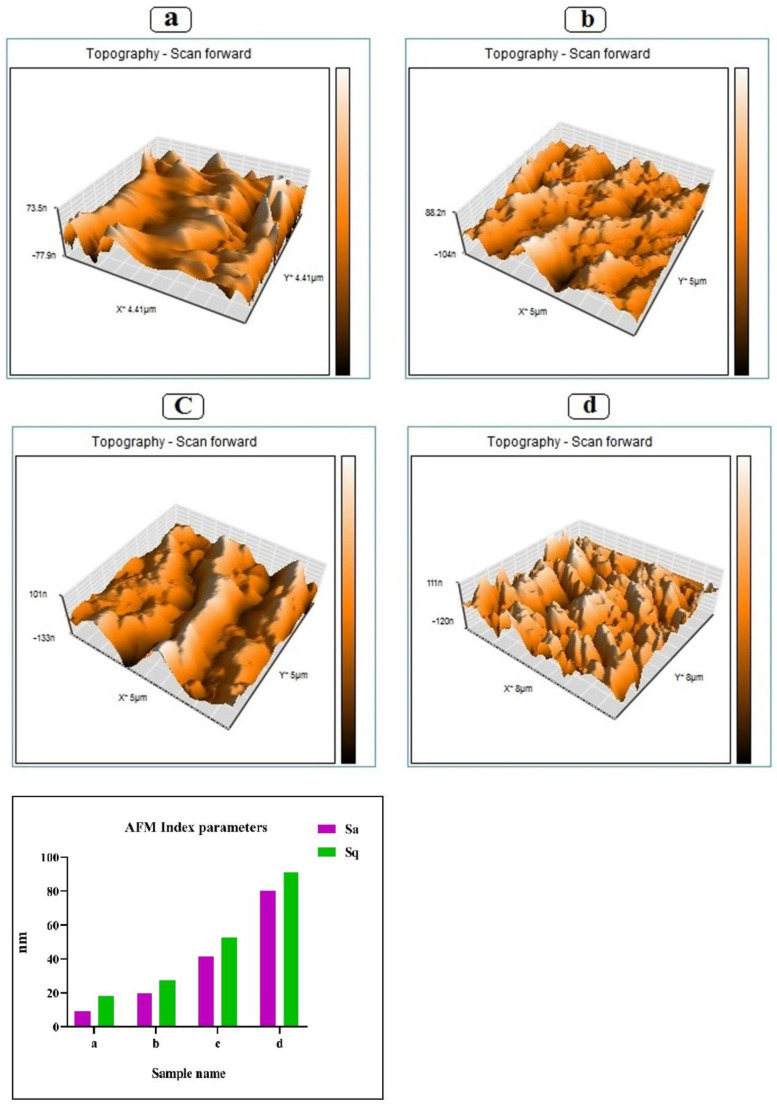
AFM 3D images and index of G/PVA (**a**), G/PVA/CWE 9 (**b**), G/PVA/MEO 9 (**c**), and G/PVA/CWE 9/MEO 9 (**d**) active films. Sa: average roughness. Sq: root mean square roughness.

**Figure 4 foods-11-03553-f004:**
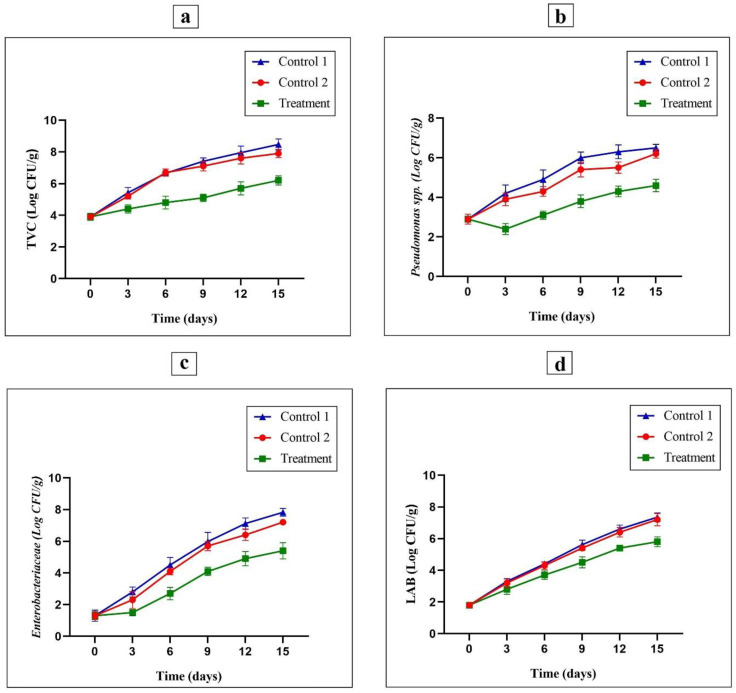
The antimicrobial effects of G/PVA/MEO/CWE films (as treatment), PE films (Control 1) and G/PVA films (Control 2) against TVC (**a**), *Pseudomonas* spp. (**b**), *Enterobacteriaceae* (**c**), and LAB counts (**d**) in in the raw turkey breast meat during 15 days of the storage in refrigeration at 4 °C.

**Figure 5 foods-11-03553-f005:**
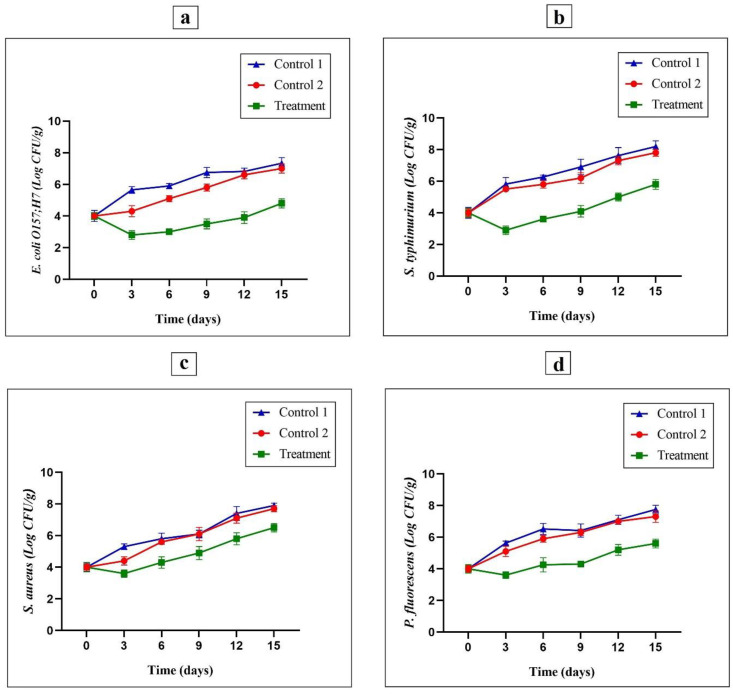
The antimicrobial effects of G/PVA/MEO/CWE films (as treatment), PE films (Control 1) and G/PVA films (Control 2) against *E. coli* (**a**), *S. typhimurium* (**b**), *S. aureus* (**c**) and *P. fluorescens* (**d**) in the inoculated raw turkey breast meat during 15 days of the storage in refrigeration at 4 °C.

**Figure 6 foods-11-03553-f006:**
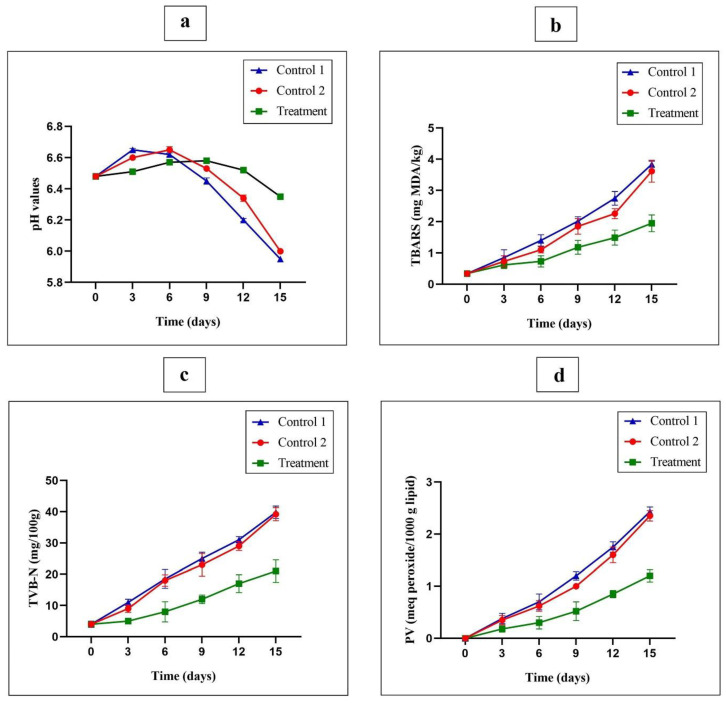
The effects of G/PVA/MEO/CWE films (as treatment), PE films (Control 1) and G/PVA films (Control 2) against the changes of pH (**a**), TBARS (**b**), TVB-N (**c**), and PV (**d**) in the raw turkey breast meat during 15 days of the storage in refrigeration at 4 °C.

**Table 1 foods-11-03553-t001:** Composition of the biodegradable active films.

CWE (mg/mL)	MEO (mg/mL)	PVA (wt%)	G (wt%)	Sample
0	0	1	2	G/PVA
3	0	1	2	G/PVA/CWE 3
6	0	1	2	G/PVA/CWE 6
9	0	1	2	G/PVA/CWE 9
0	3	1	2	G/PVA/MEO 3
0	6	1	2	G/PVA/MEO 6
0	9	1	2	G/PVA/MEO 9
3	3	1	2	G/PVA/CWE 3/MEO 3
6	3	1	2	G/PVA/ CWE 3/MEO 6
9	3	1	2	G/PVA/ CWE 3/MEO 9
3	6	1	2	G/PVA/ CWE 6/MEO 3
6	6	1	2	G/PVA/ CWE 6/MEO 6
9	6	1	2	G/PVA/ CWE 6/MEO 9
3	9	1	2	G/PVA/CWE 9/MEO 3
6	9	1	2	G/PVA/CWE 9/MEO 6
9	9	1	2	G/PVA/CWE 9/MEO 9

G: Gellan. PVA: Polyvinyl alcohol. MEO: Myrtle essential oil. CWE: Caucasian whortleberry extract.

**Table 2 foods-11-03553-t002:** Mechanical properties and thickness of the biodegradable active films.

CWE and MEO (mg /mL Film Solution)	UTS (Mpa)	SB (%)	Thickness (mm)
G/PVA	21.12 ± 0.31 ^a^	28.42 ± 0.42 ^k^	0.10 ± 0.00 ^b^
G/PVA/CWE 3	20.86 ± 0.86 ^ab^	30.65 ± 1.03 ^jk^	0.10 ± 0.00 ^b^
G/PVA/CWE 6	20.17 ± 0.52 ^abc^	33.10 ± 1.65 ^j^	0.10 ± 0.00 ^b^
G/PVA/CWE 9	19.45 ± 0.45 ^bc^	36.38 ± 0.38 ^i^	0.10 ± 0.00 ^b^
G/PVA/MEO 3	19.05 ± 0.30 ^cd^	40.94 ± 0.94 ^h^	0.10 ± 0.00 ^b^
G/PVA/MEO 6	16.66 ± 0.66 ^ef^	54.73 ± 0.51 ^f^	0.11 ± 0.00 ^ab^
G/PVA/MEO 9	12.52 ± 1.11 ^i^	67.11 ± 0.11 ^c^	0.11 ± 0.00 ^ab^
G/PVA/CWE 3/MEO 3	18.67 ± 1.40 ^cd^	41.64 ± 1.64 ^h^	0.11± 0.00 ^ab^
G/PVA/ CWE 3/MEO 6	15.25 ± 0.25 ^fg^	56.22 ± 0.44 ^ef^	0.11 ± 0.01 ^ab^
G/PVA/ CWE 3/MEO 9	11.40 ± 1.20 ^ij^	70.37 ± 0.22 ^b^	0.12 ± 0.00 ^ab^
G/PVA/ CWE 6/MEO 3	17.81 ± 0.83 ^de^	44.87 ± 0.57 ^g^	0.11 ± 0.00 ^ab^
G/PVA/ CWE 6/MEO 6	14.14 ± 0.64 ^gh^	58.79 ± 0.83 ^de^	0.12 ± 0.00 ^ab^
G/PVA/ CWE 6/MEO 9	10.27 ± 0.37 ^jk^	73.29 ± 0. 35 ^ab^	0.12 ± 0.00 ^ab^
G/PVA/CWE 9/MEO 3	16.95 ± 1.05 ^e^	46.16 ± 0.42 ^g^	0.12 ± 0.00 ^ab^
G/PVA/CWE 9/MEO 6	12.93 ± 0.93 ^hi^	61.10 ± 0.29 ^d^	0.12 ± 0.00 ^ab^
G/PVA/CWE 9/MEO 9	9.18 ± 1.06 ^k^	76.18 ± 0.65 ^a^	0.13 ± 0.00 ^a^

Different lowercase letters were determined to be significant at *p* ≤ 0.05. UTS: Ultimate tensile strength. SB: Strain at break.

**Table 3 foods-11-03553-t003:** Water vapor permeation and moisture content of the active films.

CWE and MEO (mg /mL Film Solution)	WVP × 10^−10^ (g/mhPa)	MC (%)
G/PVA	5.21 ± 0.21 ^j^	11.21 ± 0.18 ^j^
G/PVA/CWE 3	4.76 ± 0.76 ^ij^	10.48 ± 0.48 ^i^
G/PVA/CWE 6	4.32 ± 0.32 ^hi^	10.19 ± 0.34 ^i^
G/PVA/CWE 9	3.91 ± 0.41 ^h^	9.86 ± 0.66 ^i^
G/PVA/MEO 3	3.71 ± 0.21 ^gh^	9.11 ± 1.11 ^h^
G/PVA/MEO 6	3.05 ± 1.73 ^efg^	7.58 ± 0.58 ^ef^
G/PVA/MEO 9	1.93 ± 0.32 b ^cd^	5.12 ± 0.12 ^c^
G/PVA/CWE 3/MEO 3	3.23 ± 0.13 ^fg^	8.53 ± 0.53 ^gh^
G/PVA/ CWE 3/MEO 6	2.71 ± 1.30 ^ef^	7.21 ± 0.21 ^de^
G/PVA/ CWE 3/MEO 9	1.53 ± 0.67 ^bc^	4.05 ± 1.10 1 ^b^
G/PVA/ CWE 6/MEO 3	2.80 ± 0.43 ^ef^	8.02 ± 0.52 ^fg^
G/PVA/ CWE 6/MEO 6	2.35 ± 0.35 ^de^	6.54 ± 0.46 ^d^
G/PVA/ CWE 6/MEO 9	1.21 ± 0.21 ^ab^	3.19 ± 0.19 ^a^
G/PVA/CWE 9/MEO 3	2.41 ± 1.60 ^de^	7.24 ± 0.34 ^de^
G/PVA/CWE 9/MEO 6	1.84 ± 0.52 ^bcd^	5.72 ± 0.72 ^c^
G/PVA/CWE 9/MEO 9	0.79 ± 0.29 ^a^	2.51 ± 1.51 ^a^

Different lowercase letters were determined to be significant at *p* ≤ 0.05. WVP: Water vapor permeation. MC: Moisture content.

**Table 4 foods-11-03553-t004:** The MIC and MBC values obtained for the MEO and CWE against different bacteria tested.

MIC (mg/mL)					MBC(mg/mL)	
Bacteria	MEO	CWE	MO/CWE	MEO	CWE	MEO/CWE
*S. typhimurium*	6	16	9	7	23	14
*S. aureus*	6	17	10	8	24	14
*E. coli*	5	15	9	6	18	11
*P. fluorescens*	5	16	9	6	22	12

MIC: Minimum inhibition concentration. MBC: Minimum bactericidal concentration. MEO: Myrtle essential oil. CWE: Caucasian whortleberry extract. MO/CWE: Combination of MEO and CWE in a ratio of 1:1.

**Table 5 foods-11-03553-t005:** Antimicrobial activity of active film containing MEO, CWE and the combination of MEO and CWE against different bacteria.

The Diameter of Inhibition Zone (mm)					
Bacteria Strains	Control 1	Control 2	MEO	CWE	MEO/CWE
*S. typhimurium*	NI	NI	14.65 ± 0.25 ^a^	13.97 ± 0.09 ^b^	16.72 ± 0.12 ^b^
*S. aureus*	NI	NI	12.55 ± 0.05 ^c^	10.78 ± 0.11 ^d^	13.10 ± 0.31 ^d^
*E. coli*	NI	NI	15.40± 0.10 ^a^	15.94 ± 0.06 ^a^	17.22 ± 0.07 ^a^
*P. fluorescens*	NI	NI	13.81± 0.21 ^b^	12.49 ± 0.19 ^c^	14.51 ± 0.14 ^c^

NI: No inhibition. Control 1: PE film as negative control. Control 2: G/PVA film without MEO and CWE as positive control. MEO: Myrtle essential oil. CWE: Caucasian whortleberry extract. MO/CWE: Combination of MEO and CWE in a ratio of 1:1. Different lowercase letters were determined to be significant at *p* ≤ 0.05.

**Table 6 foods-11-03553-t006:** The effect of active film containing MEO and CWE on the sensory properties of turkey breast meat stored at 4 °C. Different lowercase letters were determined to be significant at *p* ≤ 0.05.

	Samples	Colour	Odor	Texture	Overall Acceptability
Days	Groups	Mean ± SD	Mean ± SD	Mean ± SD	Mean ± SD
0	Control 1 Control 2	9.00 9.00	9.00 9.00	9.00 9.00	9.00 9.00
	Treatment	9.00	9.00	9.00	9.00
3	Control 1 Control 2	7.5 ± 0.52 ^b^ 7.83 ± 0.71 ^b^	7.33 ± 0.81 ^b^ 7.50 ± 0.48 ^b^	7.16 ± 0.75 ^b^ 7.33 ± 0.61 ^b^	7.16 ± 0.51 ^b^ 7.50 ± 0.86 ^b^
	Treatment	8.83 ± 0.32 ^a^	8.50 ± 0.81 ^a^	8.33 ± 0.54 ^a^	8.50 ± 0.48 ^a^
6	Control 1 Control 2	4.83 ± 0.51^de^ 5.33 ± 0.28 ^d^	5.33 ± 0.40 ^c^ 5.83 ± 0.69 ^c^	6.16 ± 0.40 ^c^ 6.33 ± 0.78 ^c^	5.33 ± 0.38 ^c^ 5.83 ± 0.45 ^c^
	Treatment	7.83 ± 0.54 ^b^	7.16 ± 0.40 ^b^	7.50 ± 0.40 ^b^	7.50 ± 0.51 ^b^
9	Control 1 Control 2	3.83 ± 1.63 ^fg^ 4.16 ± 0.82 ^ef^	3.33 ± 0.33 ^ef^ 3.83 ± 0.28 ^de^	3.50 ± 0.54 ^efg^ 3.83 ± 0.46 ^ef^	3.83 ± 0.40 ^d^ 4.16 ± 1.05 ^d^
	Treatment	6.33 ± 0.55 ^c^	5.50 ± 0.51 ^c^	6.16 ± 0.54 ^c^	5.83 ± 0.44 ^c^
12	Control 1 Control 2	3.33 ± 2.63 ^gh^ 3.83 ± 0.71 ^fg^	2.83 ± 0.38 ^fg^ 3.16 ± 0.52 ^efg^	3.16 ± 0.40 ^f^ 3.50 ± 1.20 ^efg^	3.50 ± 0.63 ^de^ 3.83 ± 0.86 ^d^
	Treatment	5.33 ± 0.55 ^d^	5.16 ± 0.51 ^c^	4.66 ± 0.40 ^d^	5.33 ± 0.54 ^c^
15	Control 1 Control 2	2.66 ± 1.40 ^h^ 2.83 ± 0.31 ^h^	2.33 ± 0.44 ^f^ 2.50 ± 1.61 ^fg^	2.83 ± 0.73 ^f^ 2.83 ± 0.49 ^g^	2.83 ± 0.54 ^e^ 2.83 ± 0.92 ^e^
	Treatment	4.66 ± 0.67 ^de^	4.33 ± 0.51 ^d^	4.16 ± 0.40 ^de^	5.16 ± 0.51 ^c^

## Data Availability

The data presented in this study are available on request from the corresponding author.

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
