# Peer review of "The Caucasian Whortleberry Extract/Myrtle Essential Oil Loaded Active Films: Physicochemical Properties and Effects on Quality Parameters of Wrapped Turkey Breast Meat"

_foods, 2022, doi:10.3390/foods11223553_

Round 1
Reviewer 1 Report
This article is devoted to the study of the process of obtaining film with various unique properties. The article is written in an accessible language, the main results are beyond doubt. The article fits well into the subject of the journal, as it relates to the topics of chemistry and physics of food products. However, there are some points that authors should pay attention to:
1. Paragraphs "2.4.7.3." - "2.4.7.5." it is desirable to explain in them why such a technique was chosen.
2. Why did the authors choose turkey breast meat? There are plenty of other foods that this research can be done on.
3. It would be good if these films were also studied by other methods of physicochemical analysis.
4. When mixing components during preparation of films, does the chemical interaction of the components occur? If yes, then it is desirable to give the reaction equation.
5. "Preparation of G/PVA/MEO/CWE films" in this paragraph, it is also desirable to provide a visual scheme of the experiment.
Author Response
- Paragraphs "2.4.7.3." - "2.4.7.5." it is desirable to explain in them why such a technique was chosen.
We explained in the revised manuscript.
- Why did the authors choose turkey breast meat? There are plenty of other foods that this research can be done on.
Based on the review of previous studies, less research and studies had been done on turkey meat, therefore turkey meat was selected as a foodstuff for packaging.
- It would be good if these films were also studied by other methods of physicochemical analysis.
Thanks for the reviewer comment. In future research studies, the suggestion of the honorable reviewer will be considered.
- When mixing components during preparation of films, does the chemical interaction of the components occur? If yes, then it is desirable to give the reaction equation.
Based on the investigations and the results of the FT-IR test, which will be published in another article, no noticeable chemical interaction was observed.
- "Preparation of G/PVA/MEO/CWE films" in this paragraph, it is also desirable to provide a visual scheme of the experiment.
Thanks for the reviewer comment. We corrected in the revised manuscript.

Reviewer 2 Report
Bagheri et al. have investigated the effects of myrtle essential oil and Caucasian whortleberry extract on mechanical, physico-mechanical and antimicrobial properties of gellan/polyvinyl alcohol film. This is a detailed systematic study and should be of interest to the readers in the field. However, the following points need to be addressed before this paper could be accepted for publications:
1. The reference citations in the text doesn’t conform to the Food journal guidelines of numbering system.
2. There are many abbreviations are used in the article. So it is better to include a list of abbreviations.
3. References should be cited for methods described in section 2.2, 2.3, 2.4.4, 2.4.5, 2.4.6, 2.4.7.2, 2.4.7.4, 2.4.7.5.
4. In all the tables and figures, the abbreviations used should be explained in full form in the respective tables footnotes or figure captions.
5. Table 4 – does it make sense to make mention of 0.00 as SD throughout?
6. Table 5 – is it necessary to have 0.00 ± 0.00 as control values in two columns for all the bacterial strains.
7. Table 9 - is it necessary to repeat the same control values at 0 days.
8. As the reader may be lost in the too many parameters tested, it is better to provide a schematic diagram (for reader to have easy home points) just above the conclusion illustrating the observed effect/trend of treatments on various quality parameters tested.
Author Response
. The reference citations in the text doesn’t conform to the Food journal guidelines of numbering system.
We corrected in the revised manuscript.
2.There are many abbreviations are used in the article. So it is better to include a list of abbreviations.
We includded in the revised manuscript.
- References should be cited for methods described in section 2.2, 2.3, 2.4.4, 2.4.5, 2.4.6, 2.4.7.2, 2.4.7.4, 2.4.7.5.
They were cited in the revised manuscript.
- In all the tables and figures, the abbreviations used should be explained in full form in the respective tables footnotes or figure captions.
The list of abbreviations is given in the text of the article. Also, explanations are provided in the footnotes of tables and figures as much as possible.
- Table 4 – does it make sense to make mention of 0.00 as SD throughout?
We corrected in the revised manuscript.
- Table 5 – is it necessary to have 0.00 ± 0.00 as control values in two columns for all the bacterial strains.
We corrected in the revised manuscript.
- Table 9 - is it necessary to repeat the same control values at 0 days.
We corrected in the revised manuscript.
- As the reader may be lost in the too many parameters tested, it is better to provide a schematic diagram (for reader to have easy home points) just above the conclusion illustrating the observed effect/trend of treatments on various quality parameters tested.
Considering the extent of the results obtained, especially the microbial, chemical and sensory results, it seems that it is not possible to present a single schematic figure for all the results obtained.
Reviewer 3
Regarding the manuscript entitled "The Caucasian whortleberry extract / Myrtle essential oil loaded active films: physicochemical properties and effects on quality parameters of wrapped turkey breast meat’’
L15-17. Please rephrase the objectives in a better way
We corrected in the revised manuscript.
P value of significant findings should be added in the abstract
We corrected in the revised manuscript.
L28. Strong conclusions and recommendations for readers should be added at the end of the abstract
We corrected in the revised manuscript.
L75. microbial cells!!
We corrected in the revised manuscript.
L89. Hypothesis is missing
Hypothesis was added in the revised manuscript.
L108. Preparation of CWE, add ref
Reffrence was added in the revised manuscript.
L139. Each specimen tested at least 3 replicated, are three replicates enough for accurate findings? And it should be 3 replicates, grammar correction.
In order to achieve accurate results, especially in tests that have a higher percentage of errors. All the results were tested and analyzed more than 3 repetitions.
L209. turkey breast meat, here and elsewhere, the font size is different compared to the whole document
We corrected in the revised manuscript.
Statistical analysis, this is a dose-dependent study, the effect of incremental levels of the essential oil and extract should be determined to investigate which dose is ideal based on the current findings. Therefore, orthogonal polynomial contrasts (linear and quadratic effects) should be used.
Thanks for the reviewer comment. It will definitely be considered in future studies.
Tables. The actual p-value from statistical analysis should be added. Means in tables are ± what?
Duncan’s multiple range test (p≤0.05) was applied for mean comparison and determining significant difference among samples. The means were presented with standard deviation.
L312. i!!
We corrected in the revised manuscript.
In the discussion section, the authors should focus on the mechanisms of action of the active ingredients and on their conceptualization.
We completed in the revised manuscript.
The antibacterial agents such as MEO and CWE bind to the components of bacteria cells and interrupt the cytoplasmic membrane. Also, antibacterial agents change the permeability of the bacteria cell wall and affect the vital biomacromolecules, such as proteins and DNA (Zhang et al., 2016). Therefore, it can be said that the antimicrobial activity of MEO and CWE is related to the presence of phenolic compounds and chemical structures with lipophilic and hydrophobic functional groups. Antimicrobial compounds with a controlled release mechanism control the growth of microorganisms over time and thus reduce the growth gradient of microorganisms.

Reviewer 3 Report
Regarding the manuscript entitled "The Caucasian whortleberry extract / Myrtle essential oil loaded active films: physicochemical properties and effects on quality parameters of wrapped turkey breast meat’’
L15-17. Please rephrase the objectives in a better way
P value of significant findings should be added in the abstract
L28. Strong conclusions and recommendations for readers should be added at the end of the abstract
L75. microbial cells!!
L89. Hypothesis is missing
L108. Preparation of CWE, add ref
L139. Each specimen tested at least 3 replicated, are three replicates enough for accurate findings? And it should be 3 replicates, grammar correction.
L209. turkey breast meat, here and elsewhere, the font size is different compared to the whole document
Statistical analysis, this is a dose-dependent study, the effect of incremental levels of the essential oil and extract should be determined to investigate which dose is ideal based on the current findings. Therefore, orthogonal polynomial contrasts (linear and quadratic effects) should be used.
Tables. The actual p-value from statistical analysis should be added. Means in tables are ± what?
L312. i!!
In the discussion section, the authors should focus on the mechanisms of action of the active ingredients and on their conceptualization.
Conclusion what is the recommended dose of the studied substances?
Author Response
Regarding the manuscript entitled "The Caucasian whortleberry extract / Myrtle essential oil loaded active films: physicochemical properties and effects on quality parameters of wrapped turkey breast meat’’
L15-17. Please rephrase the objectives in a better way
We corrected in the revised manuscript.
P value of significant findings should be added in the abstract
We corrected in the revised manuscript.
L28. Strong conclusions and recommendations for readers should be added at the end of the abstract
We corrected in the revised manuscript.
L75. microbial cells!!
We corrected in the revised manuscript.
L89. Hypothesis is missing
Hypothesis was added in the revised manuscript.
L108. Preparation of CWE, add ref
Reffrence was added in the revised manuscript.
L139. Each specimen tested at least 3 replicated, are three replicates enough for accurate findings? And it should be 3 replicates, grammar correction.
In order to achieve accurate results, especially in tests that have a higher percentage of errors. All the results were tested and analyzed more than 3 repetitions.
L209. turkey breast meat, here and elsewhere, the font size is different compared to the whole document
We corrected in the revised manuscript.
Statistical analysis, this is a dose-dependent study, the effect of incremental levels of the essential oil and extract should be determined to investigate which dose is ideal based on the current findings. Therefore, orthogonal polynomial contrasts (linear and quadratic effects) should be used.
Thanks for the reviewer comment. It will definitely be considered in future studies.
Tables. The actual p-value from statistical analysis should be added. Means in tables are ± what?
Duncan’s multiple range test (p≤0.05) was applied for mean comparison and determining significant difference among samples. The means were presented with standard deviation.
L312. i!!
We corrected in the revised manuscript.
In the discussion section, the authors should focus on the mechanisms of action of the active ingredients and on their conceptualization.
We completed in the revised manuscript.
The antibacterial agents such as MEO and CWE bind to the components of bacteria cells and interrupt the cytoplasmic membrane. Also, antibacterial agents change the permeability of the bacteria cell wall and affect the vital biomacromolecules, such as proteins and DNA (Zhang et al., 2016). Therefore, it can be said that the antimicrobial activity of MEO and CWE is related to the presence of phenolic compounds and chemical structures with lipophilic and hydrophobic functional groups. Antimicrobial compounds with a controlled release mechanism control the growth of microorganisms over time and thus reduce the growth gradient of microorganisms.
Conclusion what is the recommended dose of the studied substances?
Based on the results obtained from the MIC and MBC tests, 14 mg/ml of essential oil and extract were selected for use in bio polymeric film matrix. Therefore, based on the positive results and the desired effect of the active films, the amount of 14 mg/ml of essential oil and extract was selected as the minimum optimal amount.

Round 2
Reviewer 3 Report
thank you for the revisions.
Author Response
there was not any comment
